# The Optimized Preparation Conditions of Cellulose Triacetate Hollow Fiber Reverse Osmosis Membrane with Response Surface Methodology

**DOI:** 10.3390/polym15173569

**Published:** 2023-08-28

**Authors:** Shu Yang, Kaikai Chen, Hongming Xiang, Yingwen Wang, Chenyan Huang

**Affiliations:** School of Textiles and Fashion, Shanghai University of Engineering and Science, Shanghai 201620, China19851543881@163.com (H.X.); wyw1975193932@163.com (Y.W.); 15716093368@163.com (C.H.)

**Keywords:** cellulose triacetate, response surface methodology, reverse osmosis, membrane, optimization

## Abstract

Reverse osmosis (RO) membrane materials play a key role in determining energy consumption. Currently, CTA is regarded as having one of the highest degrees of chlorine resistance among materials in the RO process. The hollow fiber membrane has the advantages of a large membrane surface area and a preparation process without any redundant processes. Herein, response surface methodology with Box–Behnken Design (BBD) was applied for optimizing the preparation conditions of the cellulose triacetate (CTA) hollow fiber RO membrane. There were four preparation parameters, including solid content, spinning temperature, post-treatment temperature, and post-treatment time, which could affect the permeability of the membrane significantly. In this study, the interaction between preparation parameters and permeability (permeate flux and salt rejection) was evaluated by regression equations. Regression equations can be applied to obtain the optimized preparation parameters of hollow fiber RO membranes and reasonably predict and optimize the permeability of the RO membranes. Finally, the optimized preparation conditions were solid content (44%), spinning temperature (167 °C), post-treatment temperature (79 °C), and post-treatment time (23 min), leading to a permeability of 12.029 (L·m^−2^·h^−1^) and salt rejection of 90.132%. This study of reinforced that CTA hollow fiber membrane may promote the transformation of the RO membrane industry.

## 1. Introduction

In recent years, there has been a growing concern about issues such as water scarcity and environmental pollution all over the world [1]. Due to the effects of climate change and freshwater resource pollution, the water shortage has further accelerated and is becoming an urgent problem. With the development of society and the increase in population, the shortage of fresh water has aroused the whole world’s attention [2,3]. Since there are abundant seawater resources, this may be an effective way to consider seawater desalination to compensate for and resolve the problem of freshwater scarcity. Gaining freshwater resources through appropriate desalination technology is of great strategic significance for building a resource-saving and environmentally friendly society [4].

Nowadays, RO is generally popular as the most important desalination technology [5]. In addition, it has slowly replaced traditional evaporation technology, such as multi-stage flash [6], and can combine with some new technologies such as capacitive deionization [7], electro dialysis [8], and membrane distillation [9], and is expected to lead the membrane industry in the future. Over the past few decades, the developments in RO membrane materials, membrane processes, feed solution pre-treatment, and the reduction in energy consumption have been studied as remarkable advances [10].

There is no denying that one of the key issues is the need to improve the permeate flux of membranes with high salt rejection in the RO filtration process. The permeate flux closely associates with the membrane properties according to molecular-transport models [11]. Up until now, most water treatment plants employ polyamide (PA) or its derivative RO membranes, which can provide satisfactory permeability and salt rejection [12]. However, the PA RO membranes have poor chemical stability in oxidation system. For example, free chlorine, which has a good resistance to biofouling, can weaken the PA RO membrane performance [13,14]. Meanwhile, the PA module configuration needs to be designed as spiral wound with a complex structure because of the narrow membrane area. By comparison, the hollow fiber membrane modules are easier to design, due to their large surface area and their preparation process, which is without any redundant processes [15].

Cellulose triacetate (CTA) has emerged as a way to prepare hollow fiber RO membranes, particularly for wastewater with severe biofouling, because of its good chlorine resistance [16]. Good performance of the CTA RO membrane can be obtained by manipulating preparation conditions. In the past, researchers usually used the one-factor-at-a-time experimental method, which not only consumed more time and expense but also neglected the effect of the interaction among factors [17,18]. Although the traditional orthogonal method is capable of considering a few factors at the same time, it cannot retrieve a function expression between the factors and response values, and it is difficult to find out the optimal factor combination and response value in the whole area.

The response surface method (RSM) is a collection of statistical and mathematical methods that gives an effective practical means for design optimization. It has been commonly used for experimental designs due to its advantages, such as reduction in the number of experiments that need be executed, resulting in lower reagent consumption and considerably less laboratory work [19,20,21]. In recent years, RSM has played an important role in the biotechnology. However, there has been little focus on the function of RSM in the membrane field. In previous works, Ismail and Lai [22] studied the preparation of defect-free asymmetric polysulfone membranes for gas separation through the manipulation of membrane fabrication variables using RSM. Idris et al. [23] used RSM to investigate the composition effect of the aqueous phase on the interfacial polymerization of the RO membrane. So far, no study has reported research on using RSM to optimize the preparation conditions of CTA hollow fiber RO membranes for good permeability. RSM was utilized to assess the relationship between response and independent variables, as well as to optimize the relevant condition of variables in order to predict the optimal value of the response. Box–Behnken design (BBD) is a standard RSM design, and is well suited for fitting a quadratic surface to estimate the coefficients of the response function [24,25].

In our previous study [26], we prepared reinforced CTA hollow fiber RO membranes and investigated the effects of solid content and the operating temperature on the performance of membranes. We found that the preparation conditions affected the performance of CTA reverse osmosis (RO) membranes. However, the detailed relationship between the preparation condition and the performance was not investigated. Therefore, to elucidate this relationship, we prepared a new type of CTA hollow fiber RO membrane by using RSM in this work. The interaction between preparation parameters and permeability (permeate flux and salt rejection) was evaluated by regression equations. The preparation parameters mainly included solid content, spinning temperature, post-treatment temperature, and post-treatment time. This work could provide a guide for the preparation process of an RO membrane that can obtain a good performance from hollow fiber RO membranes in practical application.

## 2. Experiments

### 2.1. Materials

CTA resins (LT35, average molecular weight (Mn) ≈ 50,000 g/mol) were purchased from Daicel (China) Investment Co., Ltd. (Shanghai, China), and benzoic acid (BA) and ethylene glycol (EG) were kindly provided by Tianjin Fengchuan Chemical Reagent Science and Technology Co., Ltd. (Tianjin, China). Tetramethylene sulfone (TMS) was obtained from Tianjin Kermel Chemical Reagent Co., Ltd. (Tianjin, China). Deionized water (DI) (pH ≈ 7.0) with a resistance of 18 MΩ was used in all experiments. NaCl was the analytical reagent and was used without further purification. Furthermore, the CTA needed to be desiccated to remove moisture in a vacuum oven (24 h, 80 ± 2 °C, 2 mbar).

### 2.2. Experimental Design

In this study, RSM was utilized to assess the relationship between the response and independent variables, as well as to optimize the relevant condition of variables in order to predict the optimal value of the response. Box–Behnken design (BBD) is a standard RSM design that is well suited for fitting a quadratic surface to estimate the coefficients of the response function [25,26]. The objective of this study was to evaluate the combined effect of preparation variables on the performance of RO membranes. The variables investigated were solid content (A), spinning temperature (B), post-treatment temperature (C), and post-treatment time (D). Permeate flux (L·m^−2^·h^−1^) and salt rejection (%) were the response variables of the RO membrane, optimized using Design-Expert software version 7.0. The levels of the factors are presented in Table 1. For each factor, a high, moderate, and low coded value was designated as 1, 0, and −1, respectively. In the experimental design twenty-nine experiment runs were carried out, which included twenty-four factorial points and five central points. Central points provide additional degrees of freedom for error estimation [27]. The experimental values and the results of the response surface are presented in Table 2. As there are only three levels for each factor, the appropriate model is the quadratic model Equation (1),
(1)$$y=\alpha_{0}+\sum_{j=1}^{n}\alpha_{j}x_{j}+\sum_{i=1}^{n}\sum_{j=1}^{n}\alpha_{ij}x_{i}x_{j}+\sum_{i=1}^{j}\alpha_{jj}x_{j}^{2}+\varepsilon$$
where *y* is the response function; *n* is the variable number; $x_{i}$ and $x_{j}$ are the values of independent variables; $a_{0},a_{j},a_{jj},a_{ij}$ are the regression coefficients for intercept, linear, quadratic, and interaction terms, respectively; ε is the error between approximate function and real function.

### 2.3. Membrane Preparation

Firstly, a ploy(p-phenyleneterephthalamide) (PPTA) twisted fiber bundle was prepared by twister with a degree of twist (20T·10 cm^−1^) and a defined mass ratio of CTA and TMS, and was homogeneously mixed under vigorous mechanical stirring. During the preparation process, TMS was regarded as a plasticizer in order to weaken the sub-valence of polymer molecules, namely van der Waals forces, as well as to increase the mobility of polymer chains. Secondly, using a twin-screw spinning machine with given parameters of temperature and according to the melt spinning method, the reinforced hollow fiber membrane with a twisted fiber bundle supporting layer was prepared. A special channel was designed for different spinneret dimensions. For the preparation process, the CTA (150 °C) melt was spun into the spinneret and coated onto the surface of PPTA twisted fiber bundle. The nascent FR CTA hollow fiber RO membranes were formed with a water coagulation bath (25 °C). Thirdly, the RO membranes were fabricated after TMS was extracted through dipping in a coagulating bath of 10 °C for 24 h. Notably, the coagulating bath was comprised by TMS/ water with a content of 35/65. In order to obtain adequate salt rejection, the RO membranes were treated with a heat treatment process for 15–45 min in water of 60–80 °C. Appendix A shows the fabrication and forming process of hollow fiber RO membranes, while the parameters and the compositions are tabulated in Table 1.

### 2.4. RO Membranes’ Permeability

It is well known that the permeability of membranes is mainly represented by permeate flux and salt rejection. In this study, the permeability of the RO membrane was investigated by a lab-scale crossflow filtration system (Appendix A). In general, 2000 mg/L NaCl aqueous solution was usually regarded as brackish water [28,29]. According to some references [30,31,32], the permeate flux and salt rejection of RO membranes could be evaluated by using NaCl aqueous solution in a lab-scale crossflow filtration system In general, the filtration system maintained the testing temperature (25 ± 2.0 °C) using a low-temperature bath circulator (RW-0525G, Lab Companion, Seoul, Korea), transmembrane pressure (2–5 MPa), and crossflow flux (1 L·min^−1^). The concentrated liquor was recycled back into the feed tank in order to maintain a constant feed concentration. Meanwhile, the conductivity of the feed solution and permeate water was obtained via a conductivity meter (DDS-11A) [33]. A glass bottle was utilized to collect permeate water and permeate flux was calculated by Equation (2).
(2)$$J_{d}=\frac{V}{A\times t}$$
where, $J_{d}$ is the permeate flux (L·m^−2^·h^−1^); *A* is the available filtration area (m^2^); *V* is the permeate water volume (L); t is the testing time (h).

A conductivity meter (DDS-11A) was utilized to measure the conductivity of the feed solution and permeate water. Salt rejection (*R*) was calculated by Equation (3).
(3)$$R=(1-\frac{C_{p}}{C_{f}})\times 100\%$$
where, $C_{p}$ is the conductivity of the permeate water; $C_{f}$ is the conductivity of the feed solution.

## 3. Results and Discussion

### 3.1. ANOVA of Permeate Flux

ANOVA was conducted, shedding light on the quality of the regression equation [34]. The analysis made use of the error sum of squares SSE (n-k-1 degrees of freedom), the total corrected sum of squares SST (n-1 degrees of freedom), and the regression sum of squares SSR (k degrees of freedom). The coefficient of determination R_2_ (R_2_ = 1 − (SSE/SST)) was a measure of the proportion of variability that is explained by the fitted model. If the model is perfect, R_2_ = 1.0. A useful analysis that determined whether a significant number of variations existed is here explained. By F-test (F = SSR/k)/(SSE/(n-k-1)), one could test the hypothesis at the a-level of significance when F > Fα (k, n-k-1). P indicated statistical significance. The statistical significance of a result was an estimated measure of the degree to which it was “true”. More technically, the value of the p-level represented a decreasing index of the reliability of a result. The higher the p-level, the less we believe that the observed relation between variables in the sample was a reliable indicator of the relation between the respective variables. Specifically, the p-level represented the probability of error that was involved in accepting our observed result as valid. In many areas of research, a p-level of 0.05 has been customarily treated as a “borderline acceptable” error level [35].

#### 3.1.1. Regression Model and Variance Analysis

The results of Table 3 were fitted with multiple regressions using Design-Expert Version 13, and the second-order fitting equation of membrane permeability flux with respect to four factors was obtained. The expression was as follows:(4)$$\begin{array}{l} J=11.76-8.38A+2.02B-0.40C+0.48D-0.80AB-0.57AC-0.28AD\\ \;\;\;\;\;\;\;\;\;+0.075BC+0.10BD-0.15CD+2.34A^{2}+0.38B^{2}+0.066C^{2}\\ \;\;\;\;\;\;\;\;\;+0.85D^{2} \end{array}$$

The quadratic polynomial regression equation corresponding to the true level of each factor was as follows:(5)$$\begin{array}{l} J=68.37312-1.80758S-0.76583T_{C}+0.92833T_{P}-0.10006t_{p}-0.004ST_{C}-\\ 0.00575ST_{C}-0.00183St_{p}+0.000375T_{C}T_{P}+0.000333T_{C}t_{p}-0.001T_{P}t_{p}+\\ 0.023408S^{2}+0.0009485T_{C}^{2}+0.0006583T_{P}^{2}+0.000379t_{P}^{2} \end{array}$$

Formula (4) was tested based on ANOVA. The variance analysis of the quadratic polynomial model is shown in Table 3. The *p* value can be seen at significant levels. The *p* < 0.0001 in this paper showed that the model had statistical significance in the scope of this experimental study. The degree of misalignment indicated the deviation between the predicted value and the experimental data. The misalignment term *p* = 0.0919 > 0.05 in this model was not significant. It showed that the model had a good fitting degree and could be used as the best preparation parameter for predicting solid content, spinning temperature, post-treatment temperature, and post-treatment time.

From Table 3, it can be seen that the F value of the model was 77.3, which meant that the model was significant. In this case, for the membrane permeation flux, A, B, A^2^, and B^2^ had a significant impact on the membrane permeation flux (*p* < 0.05). The influences of the four factors on the membrane permeate flux were ranked as follows: solid content, spinning temperature, post-treatment time, and post-treatment temperature, which could be judged by the size of F value.

The reliability of the model was analyzed and the results are shown in Table 4. The value of the complex correlation coefficient, which was closer to one, would indicate that the fitting degree of the equation was good [36]. The model was 0.9872, which showed that the fitting regression equation was reasonable and reliable. The prediction effect of the model are indicated by the modified complex correlation coefficient and the predicted complex correlation coefficient. The larger the coefficient was, the better the prediction effect obtained [37]. The coefficien of the model was greater than 0.95, indicating that more than 95% of the values could be explained by the model. Since AP = 31.1638 > 4, accidental errors had little effect on the experimental results and the model could predict the entire process accurately enough. CV = 7.02% < 10%, indicating fewer abnormal data. The binary regression equation fitted well and had high reliability. It could be used to predict the relationship between membrane permeation flux and solid content, spinning temperature, post-treatment time, and post-treatment temperature.

#### 3.1.2. Verification of the Accuracy of Membrane Permeation Flux Model

A residual is essentially the difference between mechanism and model predicted value (fitting value) [38]. It is often used to verify the hypothesis of the model, to detect outliers, and then to facilitate the modification of the model [39]. In the absence of outliers, residuals should conform to the normal distribution, and the points in the residuals’ distribution should be as straight as possible. The normal distribution of residuals in this model is shown in Figure 1A. Most of the experimental points are evenly distributed on or near the straight line, which shows that the residual obeys normal distribution with few outliers and that the fitting of the model was accurate. The comparison between the predicted and experimental values of the model is shown in Figure 1B. Most of the experimental points are on or near the straight line. The predicted values are close to or equal to the measured values, which proves that the model was suitable.

### 3.2. ANOVA of Salt Rejection

#### 3.2.1. Regression Model and Variance Analysis

Design-Expert Version 13 was utilized to fit the results of Table 2, and the second-order fitting equation of the membrane desalination rate with respect to four factors was obtained. The expression was as follows:(6)$$\begin{array}{l} J=92.26-24.44A-1.88B+1.40C-0.88D+2.38AB+1.05AC-0.35AD\\ +1.47BC-0.17BD-0.52CD-18.33A^{2}-1.21B^{2}-1.05C^{2}-2.90D^{2} \end{array}$$

Formula (6) was tested based on ANOVA. The variance analysis was shown in Table 5. From Table 5, the model *p* < 0.0001 showed that the model was extremely significant. The value *p* = 0.095 > 0.05 for the unfit item of the model was not significant, which indicated that the experimental points could be explained by the model. The A, B, C, AB, A^2^, D^2^ term in the quadratic model had a significant effect on membrane salt rejection (*p* < 0.05), and the interaction between solid content and spinning temperature was strong. From the F value, it could be judged that in the selected experimental range, the four factors affecting the salt rejection of the membrane are sorted as follows: solid content, spinning temperature, post-treatment temperature, and post-treatment time.

The reliability of the model was analyzed, and the results are shown in Table 6. The $R^{2}$ of this model was 0.9937, $R_{adj}^{2}$, $R_{pre}^{2}$, RR was greater than 0.95 and $|R_{adj}^{2}-R_{pre}^{2}|<0.05$, AP = 37.3634 > 4, CV = 2.51% < 10%. The binary regression equation could be used to predict the effects of solid content, spinning temperature, post-treatment temperature, and post-treatment time on membrane salt rejection.

#### 3.2.2. Verification of the Accuracy of the Membrane Salt Rejection Model

The normal probability distribution of the residual of the model is shown in Figure 2A, which shows that the residual distribution conformed to the normal distribution. The comparison between the predicted value and the experimental value of the desalination rate model is shown in Figure 2B, which shows that most of the experimental points fall on a straight line, and the predicted value was close to or equal to the measured value, which proves that the model was reliable.

### 3.3. The Effects of Four Variables on Permeate Flux

The permeability (permeate flux and salt rejection) could be simulated by the response surfaces data when the counter diagram showed different shapes. In general, the interaction between the parameters was not significant when the counter was circular, which can be neglected, while the factors were significant when the contour was elliptic or saddle [39,40]. In this work, the diagrams of counter and three-dimensional (3D) surface were fitted to give the interaction effects between the preparation parameters, including solid content (A), spinning temperature (B), post-treatment temperature (C), post-treatment time (D), and permeability (permeate flux and salt rejection).

The effect of solid content (A) and spinning temperature (B) on the permeate flux is shown in Figure 3-Q1. The elliptical shape of the contour plot depicts that the interaction between A and B was significant, and the impact factor of A was greater than B. It can be seen in the 3D diagram that the permeate flux obviously decreased with increasing A. This phenomenon was mainly due to the increase in the viscosity of the membrane-forming system, which made the separation layer compact [41,42]. Compared with A, B showed a relatively slight effect on the permeate flux. When B was high, the movement of molecular chains accelerated, which would have caused strong interactions among the plasticizers and molecules, thus, the tangled chains of molecules were unwinding and forming a loose structure. Meanwhile, the surface of the membrane-forming system could instantaneously reach a high polymer concentration, and the increase in polymer concentration hindered the outward diffusion of volatile plasticizer, finally forming a membrane with a thin dense layer. It was well known that the RO membranes with a thin separation layer were usually prepared with separation performance [43]. For the synthesis effects of A and B, it could be expected that while decreasing A to achieve a significant enhancement of permeate flux, at the same time a high spinning temperature would also be required.

Figure 3-Q2 shows the contour and response surface plot of solid content (A) and post-treatment temperature (C). It can be seen that there was a relatively significant interaction between A and C from observation of the elliptical nature of the contour plot. The surface plot shows that permeate flux increased with the decrease in A. The reason for this has been discussed previously. In addition, the permeate flux decreased slightly but remained basically unchanged, with C ranging between 60 and 80 (°C). This was because as the C increased, the partial binding water in the CTA molecules gained energy, overcoming the hydrogen bonding and breaking away [44]. This also increased the kinetic motion of the polymer molecular chains as well as enhancing the molecular movement, and the structure of the polymer molecular chains became more compact and more stable. Furthermore, the increase in C made the polar groups in the polymer molecular chains attract to each other, making the membrane dehydrate and shrink, which led to a reduction in permeate flux.

The response surface and the interaction of solid content (A) and post-treatment time (D) is plotted in Figure 3-Q3. According to the contour plot, the interaction between A and D was not obvious. The surface plot shows that permeate flux decreased slightly by increasing D. This was mainly due to the molecule structure of membrane surface becoming tighter with the increase in D, which made the desalination layer on the surface of the membrane more compact. Although changing D in the range of 15–45 min did not have much influence on permeate flux, decreasing A still played a significant role in the enhancement of permeate flux.

Figure 3-Q4 presents the response surface and the contour plot of spinning temperature (B) and post-treatment temperature (C). The contour plot shows that there was an interaction between B and C. It was observed that permeate flux increased by increasing B and increasing C, and the effect of B was more significant than C. As shown in Figure 3-Q5, B had no interaction with D, as is evident from the relatively circular nature of the contour curves. In addition, by increasing D, the permeate flux was obviously decreased. According to Figure 3-Q6, there was an obvious interaction effect between C and D. Moreover, the surface plot presents a similar behavior in the effect of C and D on the permeate flux.

Although all factors have influence on membrane permeate flux, the main impact factors were solid content and spinning temperature, while the others have only a minor effect.

### 3.4. Effects of Four Variables on Salt Rejection

Figure 4-K1 shows the contour and response surface plot of salt rejection as a function of solid content (A) and spinning temperature (B). The elliptical nature of the contour plot shows that the interaction between A and B was significant. It can be observed from the response surface plot that salt rejection considerably increased on increasing A, while slightly decreased on increasing B. The effect of A on salt rejection was more obvious than B. This was mainly due to the density of the membrane desalination layer increasing upon increasing A, which increased the rejection of NaCl [45]. This meant that a noticeable improvement of salt rejection could be expected by increasing solid content, even at a low spinning temperature.

The combined effect of solid content (A) and post-treatment temperature (C) on salt rejection is shown in Figure 4-K2. The contour plot shows that the interaction between A and C was relatively significant, and the salt rejection increased slightly but remained basically unchanged with C ranging between 60 and 80 (°C). Compared with A, the increase in C results in less of an impact on salt rejection. Consequently, the optimal salt rejection needed to be further considered by increasing A to a certain value to achieve a greater separation performance.

Figure 4-K3 shows the responses of salt rejection by varying solid content (A) and post-treatment time (D). This was mainly due to the surface molecular structure of the membrane becoming tighter by increasing D, which made the desalination layer on the surface of the membrane more compact [46]. Furthermore, the surface plot shape of salt rejection in Figure 4-K3 was similar to that in Figure 4-K2; the increasing of A, C, and D had the same function in salt rejection. It could be concluded that although the salt rejection could be improved by an increase in A, C, or D, the increase in A was more efficient in achieving a higher value of salt rejection.

The combined effect of spinning temperature (B) and post-treatment temperature (C) on salt rejection is presented in Figure 4-K4. The contour plot shows that there was an interaction between B and C. The three-dimensional plot shows that salt rejection decreased by increasing B, and the decrease was negligible when C was high. It is indicated that the increase in C was beneficial for achieving a high level of salt rejection. As the elliptical nature of the contour plot shown in Figure 4-K5 illustrates, the interaction between B and D was significant. The most favorable condition for high salt rejection was the combination of high B with moderate D. Figure 4-K6 shows that the increase in D had the same function as decreasing C in terms of salt rejection. The result also suggests that a high salt rejection could be obtained with a moderate D, which is similar to the conclusion from Figure 4-K5.

### 3.5. Optimization of RO Process

Response surface models were developed to predict permeate flux and salt rejection. The optimum predicted conditions, solid content 44.024%, spinning temperature 167.1 °C, post-treatment temperature 79.37 °C, and post-treatment time 23.87 min, were determined using Design-Expert. For practical application, taking the integer portion of the optimum conditions and the three tests undertaken during the experiment, the results are presented in Table 7.

It can be seen that the relative errors between the experimental and predicted values of permeate flux and salt rejection were 5.1% and 1.37%, respectively. This shows that the model fitted by response surface could reflect the real process, visually showing the influence of preparation parameters on RO performance and obtaining the best parameters for the process. It also shows that it was feasible to use RSM to optimize the formulation of RO membranes, which was beneficial for achieving a high value of salt rejection at a considerably low cost of investment and thermal energy.

## 4. Conclusions

In this work, the preparation parameters for CTA hollow fiber RO membranes were optimized based on the BBD. The permeability of the membrane could be mainly decided by solid content, spinning temperature, post-treatment temperature, and post-treatment time. The regression equations between the preparation parameters and the permeability were established by RSM, and variance analysis was carried out. The corresponding contour and three-dimensional plots were obtained as analysis for the effects of each parameter on permeate flux and salt rejection. The permeate flux was mainly determined by solid content, and salt rejection was dependent on the main effects of solid content and spinning temperature. According to the regression equations, the optimal membranes’ permeate flux of 12.029 (L·m^−2^·h^−1^) and salt rejection of 90.132% were obtained under the following conditions: solid content 44%, spinning temperature 167 °C, post-treatment temperature 79 °C, and post-treatment time 23 min. The regression model can be applied to obtain the optimal formulation of RO membranes and can reasonably predict and optimize the permeability and desalination of RO membranes. In addition, the RO performance in both production and energy efficiency can be significantly improved through this optimization. These optimal parameters may provide guidance for manufacturing the hollow fiber RO membrane in the actual production process.

## Figures and Tables

**Figure 1 polymers-15-03569-f001:**
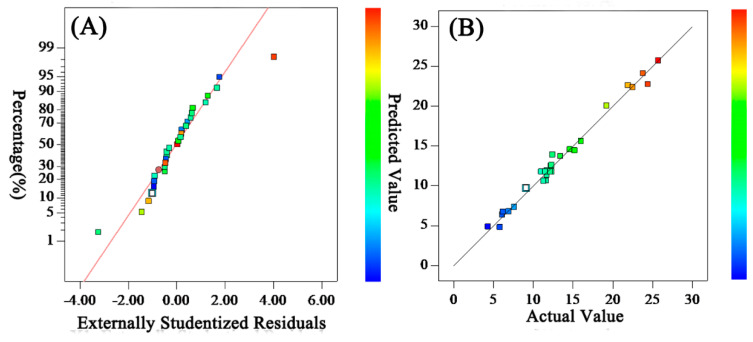
Residuals’ normal distribution (**A**) and comparison of predicted and experimental values (**B**) for Permeation Flux.

**Figure 2 polymers-15-03569-f002:**
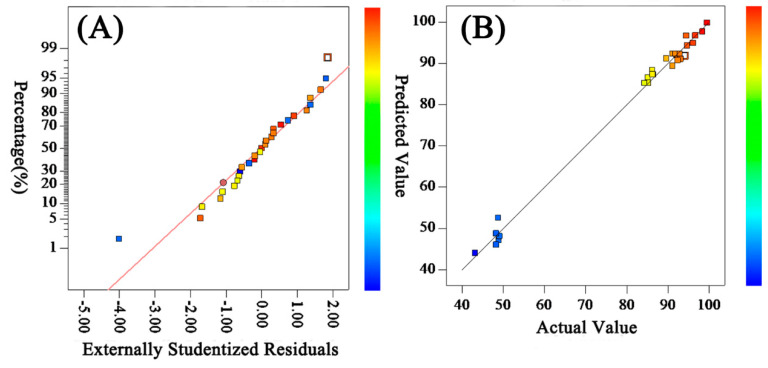
Residuals’ normal distribution (**A**) and comparison of predicted and experimental values (**B**) for salt rejection.

**Figure 3 polymers-15-03569-f003:**
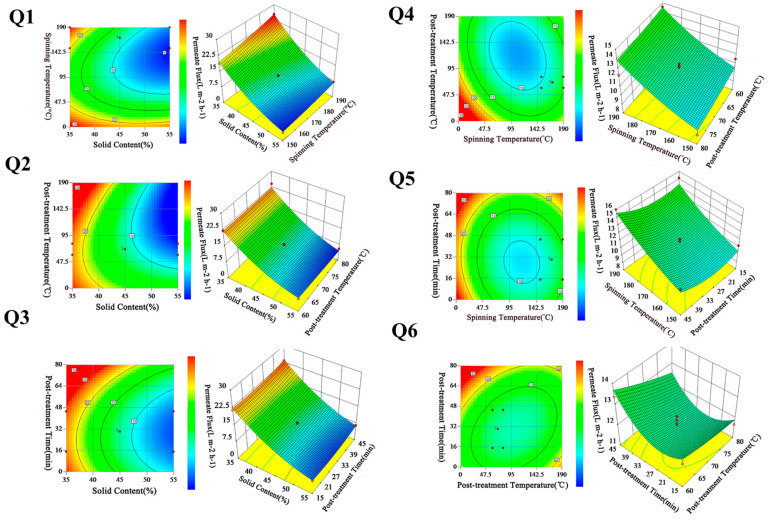
Contour and 3D diagram of different preparation parameters on permeate flux.

**Figure 4 polymers-15-03569-f004:**
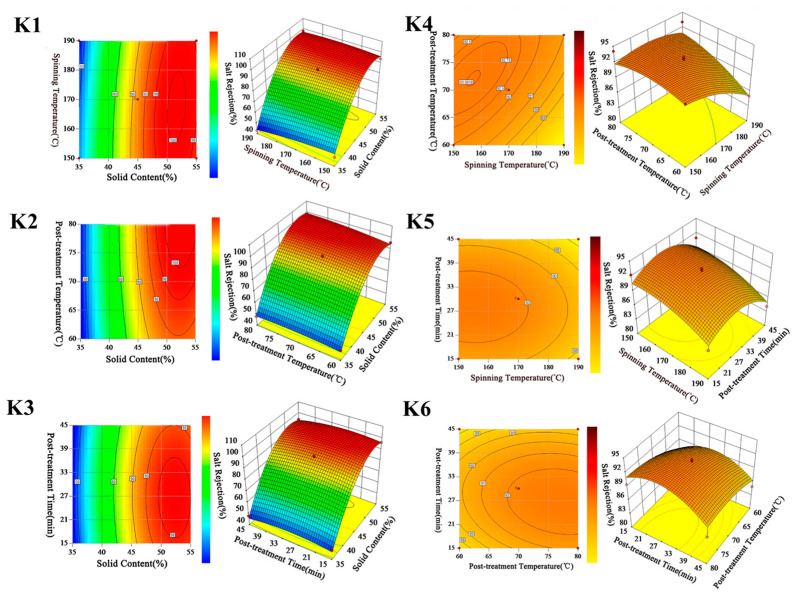
Contour and 3D diagram of different preparation parameters on permeate flux and on salt rejection.

**Table 1 polymers-15-03569-t001:** Experimental ranges and levels of the factors.

Factors	Coded Level		
	Levels of Factors		
	−1	0	+1
(A) Solid Content (%)	35.0	45.0	55.0
(B) Spinning Temperature (°C)	150.0	170.0	190.0
(C) Post-Treatment Temperature (°C)	60.0	70.0	80.0
(D) Post-Treatment Time (min)	15.0	30.0	45.0

**Table 2 polymers-15-03569-t002:** Box–Behnken design experimental design table and the corresponding permeate flux ((L·m^−2^·h^−1^)) and salt rejection (%).

Run No.	A (%)	B (°C)	C (°C)	D (min)	Permeate Flux (L·m^−2^·h^−1^)	Salt Rejection (%)
1	35.0	150.0	70.0	30.0	19.2	48.8
2	55.0	150.0	70.0	30.0	4.3	94.5
3	35.0	190.0	70.0	30.0	25.7	43.2
4	55.0	190.0	70.0	30.0	7.6	98.4
5	45.0	170.0	60.0	15.0	12.2	86.4
6	45.0	170.0	80.0	15.0	11.7	89.6
7	45.0	170.0	60.0	45.0	13.4	85.1
8	45.0	170.0	80.0	45.0	12.3	86.2
9	35.0	170.0	70.0	15.0	21.9	48.9
10	55.0	170.0	70.0	15.0	6.1	96.7
11	35.0	170.0	70.0	45.0	23.8	48.3
12	55.0	170.0	70.0	45.0	6.9	94.7
13	45.0	150.0	60.0	30.0	11.6	92.1
14	45.0	190.0	60.0	30.0	14.6	85.2
15	45.0	150.0	80.0	30.0	9.1	94.1
16	45.0	190.0	80.0	30.0	12.4	93.1
17	35.0	170.0	60.0	30.0	22.5	49.1
18	55.0	170.0	60.0	30.0	6.2	96.1
19	35.0	170.0	80.0	30.0	24.4	48.3
20	55.0	170.0	80.0	30.0	5.8	99.5
21	45.0	150.0	70.0	15.0	11.3	92.4
22	45.0	190.0	70.0	15.0	15.2	86.3
23	45.0	150.0	70.0	45.0	11.7	91.1
24	45.0	190.0	70.0	45.0	16	84.3
25	45.0	170.0	70.0	30.0	11.9	92.5
26	45.0	170.0	70.0	30.0	11	92.8
27	45.0	170.0	70.0	30.0	12.3	91.2
28	45.0	170.0	70	30	12.1	91.9
29	45.0	170.0	70	30	11.5	92.9

**Table 3 polymers-15-03569-t003:** ANOVA for regression equation of permeate flux.

Source	Sum of Squares	D_f_	Mean Square	F Value	*p*-Value Prob > F	
Model	939.6	14	67.1	77.3	<0.0001	Significant
A-Solid Content	843.4	1	843.4	971.8	<0.0001	
B-Spinning Temperature	49.2	1	49.2	56.7	<0.0001	
C-Post-Treatment Temperature	1.9	1	1.9	2.2	0.1591	
D-Post-Treatment	2.7	1	2.7	3.1	0.0991	
Time						
AB	2.5	1	2.6	2.9	0.1079	
AC	1.3	1	1.3	1.5	0.2373	
AD	0.3	1	0.3	0.3	0.5643	
BC	0.02	1	0.02	0.03	0.8744	
BD	0.04	1	0.04	0.04	0.8331	
CD	0.09	1	0.09	0.1	0.7522	
A^2^	35.5	1	35.5	40.9	<0.0001	
B^2^	0.9	1	0.9	1	0.3185	
C^2^	0.02	1	0.03	0.03	0.8597	
D^2^	4.7	1	4.7	5.4	0.0351	
Residual	12.1	14	0.9			
Lack of Fit	11.1	10	1.1	4.1	0.0919	Not Significant
Pure Error	1.1	4	0.3			
Cor Total	951.6	28				

**Table 4 polymers-15-03569-t004:** The credibility analysis of the permeate flux model.

Project	Number	Project	Number
Std. Dev.	0.93	R-Squared	0.9872
Mean	13.26	Adj R-Squared	0.9744
C.V. %	7.02	Pred R-Squared	0.9511
PRESS	65.48	Adeq Precision	31.1638

**Table 5 polymers-15-03569-t005:** ANOVA for regression equation of salt rejection.

Source	Sum of Squares	D_f_	Mean Square	F Value	*p*-Value Prob > F	
A-Solid Content	9525.3	14	680.3	158.1	<0.0001	Significant
B-Spinning Temperature	7168.7	1	7168.7	1665.7	<0.0001	
C-Post-Treatment Temperature	42.1	1	42.1	9.8	0.0074	
D-Post-Treatment	23.5	1	23.5	5.4	0.0348	
Time						
AB	9.3	1	9.3	2.2	0.1623	
AC	22.5	1	22.5	5.2	0.0381	
AD	4.4	1	4.4	1.1	0.3286	
BC	0.4	1	0.4	0.11	0.7408	
BD	8.7	1	8.7	2.1	0.1769	
CD	0.1	1	0.1	0.028	0.8684	
A^2^	1.1	1	1.1	0.26	0.6206	
B^2^	2180.3	1	2180.3	506.6	<0.0001	
C^2^	9.4	1	9.4	2.2	0.1598	
D^2^	7.1	1	7.11	1.6	0.2197	
Residual	54.4	1	54.43	12.6	0.0032	
Pure Error	58.2	10	5.82	11.5	0.0953	Significant
Cor Total	2	4	0.5			
A-Solid Content	9585.5	28				

**Table 6 polymers-15-03569-t006:** The credibility analysis of the model of salt rejection.

Project	Number	Project	Number
Std. Dev.	2.07	R-Squared	0.9937
Mean	82.54	Adj R-Squared	0.9874
C.V. %	2.51	Pred R-Squared	0.9646
PRESS	338.60135	Adeq Precision	37.3634

**Table 7 polymers-15-03569-t007:** Optimized experimental results.

No.	Permeate Flux (L·m^−2^·h^−1^)	Salt Rejection%
Experimental Group 1	12.7	92.1
Experimental Group 2	11.6	93.5
Experimental Group 3	13.2	86.9
Predicted Value	12.03	90.13
Relative Error	3.77%	0.79%

## Data Availability

Not applicable.

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
