# Peer review of "The Optimized Preparation Conditions of Cellulose Triacetate Hollow Fiber Reverse Osmosis Membrane with Response Surface Methodology"

_polymers, 2023, doi:10.3390/polym15173569_

Round 1

Reviewer 1 Report

Dear Editor,

The manuscript titled "The optimized preparation conditions of cellulose triacetate hollow fiber reverse osmosis membrane with response surface methodology" presents an investigation into the optimization of preparation conditions for cellulose triacetate (CTA) hollow fiber reverse osmosis (RO) membranes using response surface methodology (RSM) with Box-Behnken Design (BBD). The authors have explored the effects of four variables (solid content, spinning temperature, post-treatment temperature, and post-treatment time) on the performance of RO membranes and established regression equations to predict membrane performance. Therefore, a major revision is recommended.

* The authors should provide a clearer justification for studying CTA membranes, explain their importance in water purification, and highlight the relevance of response surface methodology for optimizing membrane preparation conditions. 

* The lack of information on error analysis and uncertainty estimation should also be addressed.

*The manuscript would benefit from the inclusion of graphical representations, such as response surface plots, to visually depict the relationship between variables and membrane performance.

*The authors should emphasize the practical implications of their findings, such as the potential for improved membrane fabrication processes in water treatment applications. The implications of the optimized preparation conditions for industrial-scale membrane production and their relevance to addressing current challenges in reverse osmosis technology should be highlighted.

* The language and writing style need improvement to enhance readability. 

( We established the regression equations between the preparation parameters and the performances of the RO membranes. Solid content, spinning temperature, post-treatment temperature and post-treatment time were considered as dominant preparation parameters in controlling performance. Main effects, quadratic effects and interactions of the four variables on the permeate flux and salt rejection were investigated.

*Some sentences require further clarity for better understanding.

In this study, we established the regression equations between the variables and the performance of the RO membranes.

Firstly, ploy(p-phenyleneterephthalamide)(PPTA) twisted fiber bundle was prepared by twister with a degree of twist(20T·10cm-1), and according to the solid content

from Table 1, a defined mass ratio of CTA and TMS was homogeneously mixed under

vigorous mechanical stirring.

Given the potential significance of the study's findings for membrane technology, I recommend a major revision of the manuscript. The authors should address the major comments provided above and carefully revise the manuscript to enhance its clarity, scientific rigor, and overall impact. Please feel free to contact me if you have any further questions or require additional clarification.

Sincerely,

Dear Editor,

The manuscript titled "The optimized preparation conditions of cellulose triacetate hollow fiber reverse osmosis membrane with response surface methodology" presents an investigation into the optimization of preparation conditions for cellulose triacetate (CTA) hollow fiber reverse osmosis (RO) membranes using response surface methodology (RSM) with Box-Behnken Design (BBD). The authors have explored the effects of four variables (solid content, spinning temperature, post-treatment temperature, and post-treatment time) on the performance of RO membranes and established regression equations to predict membrane performance. Therefore, a major revision is recommended.

* The authors should provide a clearer justification for studying CTA membranes, explain their importance in water purification, and highlight the relevance of response surface methodology for optimizing membrane preparation conditions. 

* The lack of information on error analysis and uncertainty estimation should also be addressed.

*The manuscript would benefit from the inclusion of graphical representations, such as response surface plots, to visually depict the relationship between variables and membrane performance.

*The authors should emphasize the practical implications of their findings, such as the potential for improved membrane fabrication processes in water treatment applications. The implications of the optimized preparation conditions for industrial-scale membrane production and their relevance to addressing current challenges in reverse osmosis technology should be highlighted.

* The language and writing style need improvement to enhance readability. 

( We established the regression equations between the preparation parameters and the performances of the RO membranes. Solid content, spinning temperature, post-treatment temperature and post-treatment time were considered as dominant preparation parameters in controlling performance. Main effects, quadratic effects and interactions of the four variables on the permeate flux and salt rejection were investigated.

*Some sentences require further clarity for better understanding.

In this study, we established the regression equations between the variables and the performance of the RO membranes.

Firstly, ploy(p-phenyleneterephthalamide)(PPTA) twisted fiber bundle was prepared by twister with a degree of twist(20T·10cm-1), and according to the solid content

from Table 1, a defined mass ratio of CTA and TMS was homogeneously mixed under

vigorous mechanical stirring.

Given the potential significance of the study's findings for membrane technology, I recommend a major revision of the manuscript. The authors should address the major comments provided above and carefully revise the manuscript to enhance its clarity, scientific rigor, and overall impact. Please feel free to contact me if you have any further questions or require additional clarification.

Sincerely,

Reviewer 2 Report

we added in the attachment the reviewer sheet.

english have minor error.

Reviewer 3 Report

Several factors during the processing of the cellulose triacetate hollow fiber reverse osmosis membrane were investigated, overall, the current version of the manuscript is a pretty rough draft with quite a few typos. The manuscript were loaded with figures without sufficient discussion/explanation why the membranes behave the way they are, and even though in the text authors refer to different panels in the figure as A, B, C or D, but I don’t see any A, B, C, D in each figures presented in the figure. The missing labels of the figures made it impossible for me to follow the text/figures described in the manuscript.

Minor:

Typo: whcih in the abstract

Keep all the decimal places consistent for the parameters across the entire manuscript, For example, Temperature and time, in section 3.5, I don’t think.three decimal places are necessary, like in Table 2, the temp and time don’t have that many decimal places. Same thing with the parameters in all the formula of J. Same thing in the abstract, for the percentage.

Table 1, in the title is ‘levers’, but in the title is ‘levels’ of factors.

Major:

From a series of contour and 3D view of the correlation between the slat rejection percentage and various processing conditions, e.g.  post-treatment time, temperature, spinning temperature solid content etc., The salt rejection percentages were optimized in individual pairs within each processing condition, however, will the salt-rejection percentage still remain the highest when combining the optimized condition from each individual conditions or if there will be any trade-off?

Since the BBD of response surface methodology was the major methods used in this manuscript, if the introduction of this methodology can be moved to the introduction instead of burying one sentence in the experimental design.

What are the reasons that these parameters were kept within the range showing in Table 1? Any experimental or experience values supporting these?

On Page 2 the previous study the authors mentioned was not referenced, which should be properly referenced, since this is a continuity study. The last sentence in the introduction only has half of the parenthesis.

Typos and non-scientific descriptions need to be eliminated.

Round 2

Reviewer 1 Report

Dear Editor,

 I am writing to formally recommend the acceptance of the revised manuscript titled " The optimized preparation conditions of cellulose triacetate hollow fiber reverse osmosis membrane with response surface methodology " following its revision by the authors in response to the initial review process.

The revisions made have significantly improved the quality and clarity of the manuscript, and I believe the study now makes a valuable contribution to the field.

Thank you for considering my recommendation. I trust that the publication of this manuscript will enrich the journal and contribute to advancing the scientific discourse in the relevant field.

Best

Reviewer 2 Report

After carefully check the revised manuscript and follow the answers, and respond to reviewers, I recomment the present manuscript to be accepted.

Reviewer 3 Report

Most of the comments were addressed by the authors, ok to be accepted. 

Quality of English is ok.